# A hybrid of light-field and light-sheet imaging to study myocardial function and intracardiac blood flow during zebrafish development

Zhaoqiang Wang[1], Yichen Ding[2], Sandro Satta[2], Mehrdad Roustaei[1], Peng Fei[3]*, Tzung K. Hsiai[1,2,4]*

**1** Department of Bioengineering, University of California, Los Angeles, California, United States of America,
**2** Division of Cardiology, Department of Medicine, School of Medicine, University of California, Los Angeles,
California, United States of America, **3** School of Optical and Electronic Information-Wuhan National
Laboratory for Optoelectronics, Huazhong University of Science and Technology, Wuhan, China,
**4** Department of Medicine, Greater Los Angeles VA Healthcare System, Los Angeles, California, United
States of America

* feipeng@hust.edu.cn (PF); thsiai@mednet.ucla.edu (TKH)

**Data Availability Statement:** The code for image post-processing, quantification and visualization is available at https://github.com/aaronzq/cardiac. The datasets generated during this study are

## Abstract

Biomechanical forces intimately contribute to cardiac morphogenesis. However, volumetric imaging to investigate the cardiac mechanics with high temporal and spatial resolution remains an imaging challenge. We hereby integrated light-field microscopy (LFM) with light-sheet fluorescence microscopy (LSFM), coupled with a retrospective gating method, to simultaneously access myocardial contraction and intracardiac blood flow at 200 volumes per second. While LSFM allows for the reconstruction of the myocardial function, LFM enables instantaneous acquisition of the intracardiac blood cells traversing across the valves. We further adopted deformable image registration to quantify the ventricular wall displacement and particle tracking velocimetry to monitor intracardiac blood flow. The integration of LFM and LSFM enabled the time-dependent tracking of the individual blood cells and the differential rates of segmental wall displacement during a cardiac cycle. Taken together, we demonstrated a hybrid system, coupled with our image analysis pipeline, to simultaneously capture the myocardial wall motion with intracardiac blood flow during cardiac development.

## Author summary

During the conception of the heart, cardiac muscular contraction and blood flow generate biomechanical forces to influence the functional and structural development. To elucidate the underlying biomechanical mechanisms, we have embraced the zebrafish system for the ease of genetic and pharmacological manipulations and its rapidity for organ development. However, acquiring the dynamic processes (space + time domain) in the small beating zebrafish heart remains a challenge. In the presence of a rapid heartbeat, microscopy is confined by temporal resolution to image the cardiac contraction and blood flow. In this context, we demonstrated an integrated light-sheet and light-field imaging system to

publicly available at https://doi.org/10.6084/m9.
figshare.14813628.v1.

**Funding:** This project was supported by NIH
R01HL111437 (T.K.H.), R01HL118650 (T.K.H.),
R01HL149808 (T.K.H.), and R00HL148493 (Y.D.).
Funder: National Institutes of Health https://www.
nih.gov/ The funders had no role in study design,
data collection and analysis, decision to publish, or
preparation of the manuscript.

**Competing interests:** The authors have declared
that no competing interests exist.

visualize cardiac contraction along with the flowing blood cells inside the cardiac chambers. Assuming the periodicity of the cardiac cycle, we synchronized the image data in post-processing for 3-D reconstruction. We further quantified the velocity of the various regions of cardiac muscular contraction, and tracked the individual blood cells during the cardiac cycles. The time-dependent velocity maps allow for uncovering differential segments of cardiac contraction and relaxation, and for revealing the patterns of blood flow. Thus, our integrated light-sheet and light-field imaging system provides an experimental basis to further investigate cardiac function and development.

This is a *PLOS Computational Biology* Methods paper.

## Introduction

The zebrafish (*Danio rerio*) model is a well-established genetic system to study cardiovascular development and disease [1–3]. During the embryonic stages, zebrafish are optically transparent, and their brief gestational period and viability in the absence of blood circulation render zebrafish a widely adopted model to study developmental biomechanics [4,5]. During cardiac morphogenesis, peristaltic contraction of the heart tube generates wall shear stress (WSS) to modulate mechanotransduction underlying cardiac looping and the initiation of endocardial trabeculation [6–8]. Both myocardial contractility and hemodynamic shear stress are essential for the formation of atrioventricular (AV) and ventriculobulbar (VB) valves [9–11]. While the molecular mechanisms underlying cardiac morphogenesis have been extensively studied [12–14], the mechanotransduction underlying myocardial contraction and hemodynamics to modulate cardiac morphogenesis have remained elusive.

The advent of *in vivo* microscopy has enabled real-time imaging of the cardiac structure and function [15,16]. Advanced imaging modalities, including confocal microscopy and light sheet fluorescence microscopy (LSFM), have allowed for capturing 3-D cardiovascular phenotypes during development with enhanced spatial and temporal resolution [17–19]. However, acquiring the dynamic events at rapid heart rates (120–180 beats per minute) still constitutes an imaging challenge using the existing techniques. Previous investigations have reported a retrospective gating method [20,21] to re-align the asynchronous time-lapse images that were sequentially acquired at each slice of the heart. This approach has been effective in reconstructing the myocardial deformation and electrical conduction [7,20,22]. However, this method is limited from capturing the intracardiac blood cells, because the periodicity assumption fails due to the random cell motion [20]. For these reasons, a majority of analyses for blood flow have adopted the 2-D image-based techniques or *in silico* models [23–26].

To capture the synchronized myocardial contraction and intracardiac hemodynamics, we integrated LSFM with light field microscopy (LFM), followed by a modified retrospective gating method. Unlike the conventional scanning imaging, the LFM captures both the lateral and angular events from the incident light in a single snapshot; thus, enabling instantaneous volumetric imaging via post-processing [27,28]. LFM has demonstrated the capacity to capture the sparse dynamic signals such as red blood cells [29–31]. Using transgenic zebrafish labeling cardiac myosin light chain, *cmlc2*, with GFP and flowing blood cells, *gata1a*, with dsRed, we were able to capture the myocardial displacement, along with intracardiac blood flow at 200 volumes per second (vps, defined as the number of acquired stacks of 2-D cross-sections of the sample per second) from 3 to 5 days post fertilization (dpf). These imaging data enabled us to extrapolate the vector maps, establishing segmental variations in the myocardial displacement

velocity and trajectories of individual blood cells traversing the ventricle during a cardiac cycle. We further revealed flow reversal during the early-stage of valve formation. The hybrid of LSFM and LFM is synergistic to couple with the retrospective gating method, allowing for acquiring simultaneous myocardial contraction and intracardiac blood flow.

## Results

### Integration of light-sheet and light-field imaging to capture myocardial displacement and intracardiac flow

We demonstrated a sequential imaging pipeline along with a modified retrospective gating method for hybrid light-field and light-sheet imaging (**Materials and methods**, **Figs 1A** and **S1** and **S2**). Using the LFM for capturing the traveling blood cells, we acquired light fields at up to 200 frames per second (fps). The heart was selectively illuminated by a rod-shaped laser beam to eliminate background noise and to enhance image contrast (**Fig 1A: upper panel**) [30]. We adopted a deep-learning algorithm [31] for 3-D reconstruction of the blood cells acquired from the raw 2-D light-field sequences. This algorithm reconstructed an equivalent 3-D imaging speed at 200 vps. Using the LSFM for imaging cardiac contraction, we captured the cross section of beating myocardium with high spatial resolution and signal-to-noise ratio (SNR). At each scanning step, we acquired the time sequence at the same frame rate as LFM at 200 fps. By adopting a modified retrospective gating method [7,20–22], we synchronized all the sequences from both LSFM and LFMs at the identical cardiac phases, and we performed 3-D reconstruction of the contracting myocardium in synchrony with the blood flow (**Fig 1A: lower panel, Fig 1B–1D and S1 Movie**). We demonstrated the atrium and ventricle, along with the intracardiac blood cells, during a cardiac cycle in the *Tg(cmlc2:GFP; gata1a:dsRed)* transgenic line from 3 to 5 dpf (**S2 Movie**). Thus, the reconstructed dual-channel zebrafish heart enabled dynamic and simultaneous analyses of the myocardial displacement and blood flow.

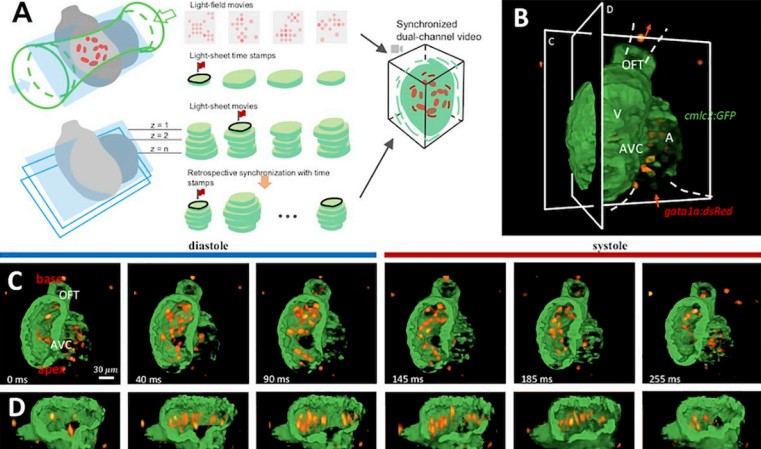

**Fig 1. A Pipeline for high speed imaging for ventricular contraction and intracardiac flow dynamics.** (A) The integration of light-sheet and light-field microscopy captures the contracting myocardium (*cmlc2:GFP*) and traveling blood cells (*gata1a:dsRed*) at 200 volumes per second. The light field-generated blood cell images (upper illustration) are synchronized with the light sheet-generated myocardial cross-section images (lower illustration). (B) GFP-labeled cardiomyocyte light chain *(cmlc2)* and dsRed-labeled blood cells (*gata1a*) are simultaneously visualized in a 3-D reconstructed embryonic heart. The red arrows indicate the direction of blood flow. A: atrium; V: ventricle. (C & D) A time sequence of a cardiac cycle is illustrated in the coronal (C) and sagittal (D) plane, respectively. During diastole (light-blue bar), the blood cells traverse the atrioventricular canal (AVC) into ventricle. During systole, the blood cells travel through the outflow tract (OFT).

### Post-imaging computation to demonstrate 4-D (space + time) vector fields for myocardial displacement and intracardiac blood flow

Images acquired by our pipeline embraced the time- and position-dependent information for vector analysis of the myocardial displacement (**Fig 2A**). We computed the myocardial displacement between two consecutive frames by using the deformable image registration (DIR) to infer the voxel-based vector field (**Fig 2B**). Each vector, divided by the time interval, indicated the direction and magnitude of myocardial velocity ($\mu m/s$). A heatmap could reveal the spatial variations of the myocardial velocity (**Fig 2C**). The increase in velocity toward the apex and atrioventricular canal (AVC) suggested a large displacement and myocardial kinetic energy, as previously reported [32]. This image-based analysis provided a robust platform to demonstrate the 3-D myocardial motion in an unsupervised manner, while the previous methods (e.g. strain analysis, fractional shortening) often involved human-annotation [7,11]. Therefore, the result acquired from our pipeline reduced inter-observer variations and increased experimental reproducibility.

To address the intracardiac blood flow, we employed the Particle Tracking Velocimetry (PTV) to track the individual cells during two cardiac cycles. We mapped the trajectories of 81 blood cells traversing from AVC to the outflow tract (OFT) (**Fig 2D**), and extracted the velocity vectors of the individual cells frame-by-frame (**Fig 2E**). We integrated DIR with PTV techniques to merge the vectors for myocardial velocity (**Fig 2B**) and the traveling blood cells (**Fig 2E**), capturing the synchronized myocardial contration with intracardiac hemodynamics (**Fig 2F** and **S3 Movie**).

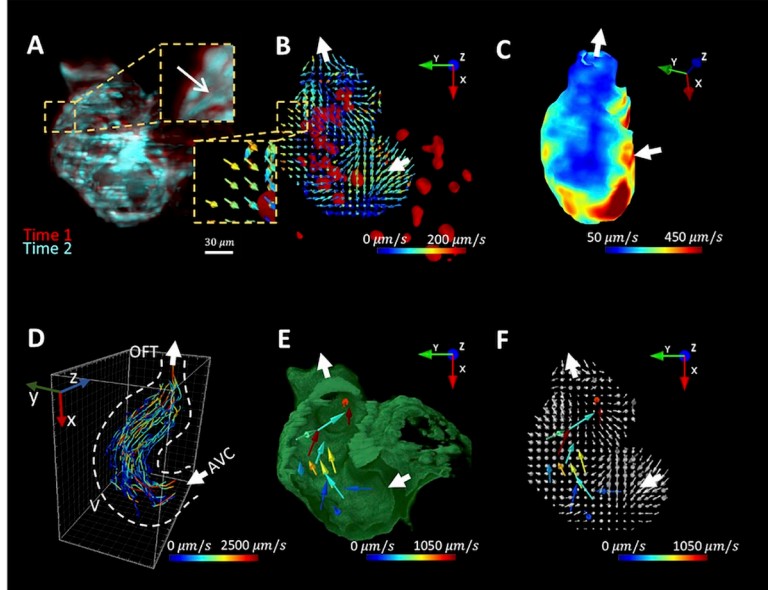

**Fig 2. Post-imaging computation to reconstruct 3-D vector fields for myocardial displacement and blood flow.**
(A) Images of the myocardium from 2 different time points are overlaid to demonstrate the myocardial displacement. The arrow indicates the direction of displacement. (B) Deformable image registration (DIR) is used to infer the myocardial displacement between frames. Each vector indicates the direction and magnitude of the displacement velocity as color-coded by the magnitude ($\mu m/s$) in the corresponding voxel. Intracardiac blood cells (red) and flow direction (white arrow) are superimposed with the vector field. (C) A representative heatmap depicts the segmental variations in the magnitude of the myocardial velocity. (D) Particle tracking velocity follows the trajectories of the blood cells during the cardiac cycles. V: ventricle; OFT: outflow tract; AVC: atrioventricular canal. (E) The vector field represents the direction and speed of the blood cells. Dissected myocardium (green) and flow direction (white arrow) are superimposed with the vector field. (F) The velocity vector fields for myocardial displacement and the traveling blood cells are merged.

## Frame-to-frame analyses of the 4-D ventricular contractility and intracardiac blood flow

To quantify the flow pattern, we defined the flow direction along the centerline that was fitted to the distribution of the traveling blood cells (**Figs 3A and S3 and S4**). We projected the velocity vectors along the flow centerline to generate a mean velocity (**Fig 3C: upper panel**). The negative values ($|\vec{V}| > 1000\ \mu m/s$) during the end of systole and diastole represented flow reversal across the valves, resulting in a decrease in the net forward flow at 3 days post fertilization (dpf). The regurgitation was reported to be a biomechanical cue to valvulogenesis [33].

The ventricle was divided into 4 segments with reference to the flow centerline to compare with the myocardial velocity in the individual segment (**Fig 3B**). We calculated the magnitude-weighted mean vectors for each segment (**Fig 3C: lower panel, color lines**), as compared with the global average magnitude (**black dotted line**). These myocardial vectors quantified the spatial variations in myocardial motion during a cardiac cycle. Using the heatmap and velocity field, we demonstrated the time-dependent myocardial velocity. We further revealed that the myocardial velocity vectors in each segment rotated clockwise or counterclockwise during a cardiac cycle (**Fig 3D**).

At 3 dpf, we observed similar temporal profiles in the blood flow velocity among several zebrafish (**S4 Movie**), where blood flow was steady and slow during systole, but instantaneous and rapid during diastole (**Fig 4A**) in synchrony with the myocardial velocity (**Fig 4B and 4C**). To minimize the fish-to-fish variation, we normalized the segmental velocity of the

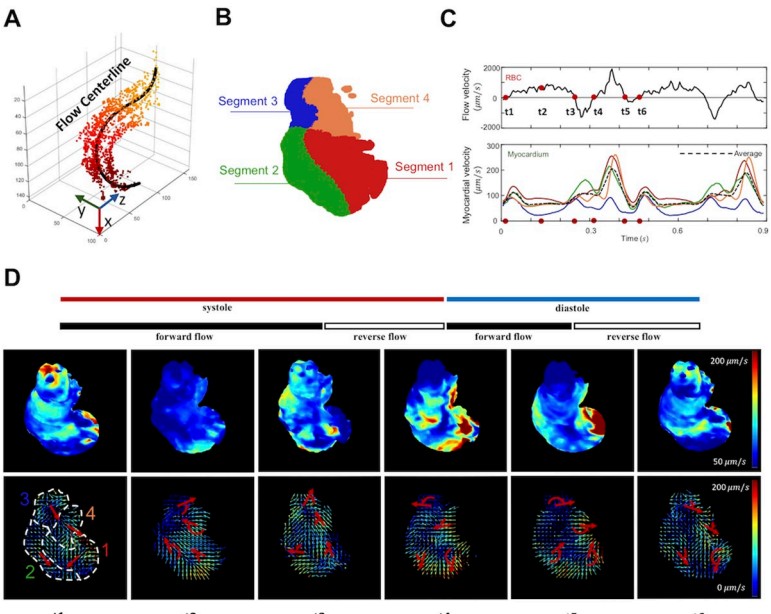

**Fig 3. Frame-to-frame analyses of the myocardial contraction and intracardiac blood flow at 3 dpf.** (A) The trajectory of the blood cells is depicted as the flow centerline, starting from the AV canal (dark colors) to the outflow tract (light colors). (B) The myocardium is divided into 4 segments to assess regional variations in the vector field. (C) The velocity of individual blood cells is projected onto the flow centerline, and the mean flow velocity is plotted as a function of time. Myocardial velocity in each segment is quantified as the magnitude-weighted mean vector for each segment, as plotted in the color line. The black dotted line demonstrates the average velocity magnitude of the entire ventricle. (D) Six representative time points (from $t_1$ to $t_6$) of the heatmaps are presented for the myocardium (upper panels, color-coded in vector magnitude) and vector maps (lower panels). Red arrows indicate the rotational direction of the mean vector for each segment.

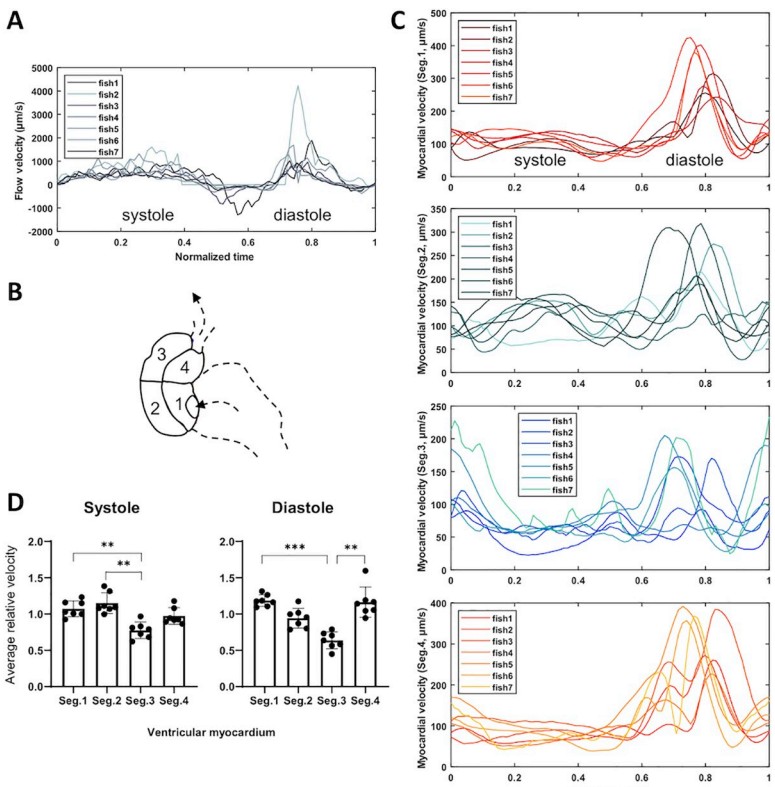

**Fig 4. Velocity profiles in myocardial displacement and intracardiac blood flow at 3 dpf.** (A) Blood flow velocity is depicted during a cardiac cycle. To compare among different heart rates, we normalize the time to one cardiac cycle for each zebrafish embryo. (B) The ventricle is divided into 4 segments. Arrows indicate the flow direction. The dotted line delineates the atrium. (C) The tracing of the myocardial velocity is normalized to one cardiac cycle in each segment. (D) The mean of relative displacement velocity is depicted in each segment during systole and diastole. The relative velocity in each segment is compared as the ratio of the segmental velocity to the mean of global velocity during a cardiac cycle. Kruskal-Wallis test and Dunn's multiple comparisons test are performed in GraphPad Prism (** $p \leq 0.01$; *** $p \leq 0.001$, n = 7).

myocardium to the global velocity for the individual zebrafish hearts during systole and diastole. We observed that myocardial contraction (systole) was more prominent in segments 1 and 2 (near the apex) as compared to segments 3 and 4, whereas myocardial relaxation (diastole) was more prominent in segments 1 and 4 (near the atrium) as compared to segments 2 and 3 (**Fig 4D**).

## Discussion

Myocardial contraction and intracardiac hemodynamics are intrinsically linked during cardiac morphogenesis. However, the contribution of hemodynamic forces during cardiac development, injury, and repair has remained poorly understood in the setting of genetic, molecular, and epigenetic factors [34]. Numerous studies have supported the notion that irregular heartbeat or obstruction in blood flow contributes to developmental defect leading to congenital heart disease [33,35,36]. In this context, we demonstrated the synchronized myocardial motion with intracardiac blood flow by integrating LFM with LSFM, coupled with the modified retrospective gating method for both periodic and irregular cardiac cycles to capture the zebrafish model of cardiac development from 3 to 5 dpf (**S2 Movie**). We identified flow reversal at the early stage of cardiac development, and also observed rapid myocardial displacement

and fast blood flow during diastole, but steady flow during systole. We further demonstrated prominent myocardial contraction near the apex, but relaxation adjacent to the atrioventricular canal. These segmental variations resonated with the image data (**S1, S2 and S4 Movies**) in which ventricular contraction propagated from the apex and relaxation propagated in a rotational manner toward the atrium.

The transgenic *Tg(cmlc2:GFP; gata1a:dsRed)* zebrafish provided the tracers of fluorescence-labeled cardiac myosin light chain and blood cells for this study. In addition to interrogating myocardial motion and blood flow, the hybrid of light-sheet and light-field imaging systems have the capacity to investigate myocardial calcium flux during electromechanical coupling [22] and intracardiac CD41-labeled GFP for platelets [37]. Furthermore, the combination of LSFM and LFM would enhance our capacity to acquire the time-lapse images for lineage tracing as the cardiac stem cells migrate from the neural crest to the developing heart tube [37]. Thus, the novelty of our hybrid imaging system coupled with the retrospective gating method provides the experimental basis to investigate segmental myocardial velocity and flow dynamics during cardiac morphogenesis.

The strength of our LFM was to provide volumetric imaging for PTV. Similar to Particle Image Velocimetry (PIV) [6,26,38], PTV has been demonstrated to track the trajectory of dynamic tracers [4]. While PIV is ineffective in tackling the sparse signals in the small space [23], PTV has the capacity to address the size of the tracer comparable to the diameter of the vessel. PTV quantifies and tracks the individual signals from the dsRed-labeled blood cells in a time-dependent manner; thereby, allowing for tracking multiple blood cells traversing from atrioventricular canal (AVC) to outflow tract (OFT) (**Fig 2**). PTV allows for validation of *in silico* simulation of intracardiac blood flow [8,23,25].

In parallel with tracking the trajectories of individual blood cells, we used the deformable image registration (DIR) to quantify the myocardial velocity in 4 different segments of the heart. Currently, fractional shortening (FS) is calculated from the changes in the diameter of the ventricular cross-section during a cardiac cycle; however, FS may underestimate cardiac function from various imaging angles [39]. Speckle tracking is another method, but it is limited by the out-of-plane motion [33]. While ejection fraction (EF) remains the gold standard for the global ventricular function, changes in volume usually require 3-D manual segmentation. In contrast, DIR can be applied to the 3-D image data for high resolution displacement map under an unsupervised environment. While the image registration may not correspond to the theoretical myocardial deformation, a close correlation was still observed in line with the results directly inferred from the 2-D kymographs, and it obviated the angle-dependent errors (**S5 Fig**). Furthermore, the velocity gradient could be quantified for the rate of local ventricular deformation (**S6 Fig**).

While performing high speed volumetric imaging remains a challenge for the current optic hardware, there exist solutions to accelerate the conventional methods for scanning imaging. However, these methods often require special optical designs plus high speed sCMOS cameras [20,40]. Our approach adopts the basic form of the existing LFM and LSFM (**S2 Fig**), alleviating the hardware burden via image post-processing. In this context, our hybrid imaging system adds a new dimension to advance the field of developmental dynamics.

## Materials and methods

### Ethics statement

All animal studies were performed in compliance with the IACUC protocol approved by the UCLA Office of Animal Research.

## Zebrafish line

The transgenic *Tg(cmlc2:GFP; gata1a:dsRed)* zebrafish (*Danio rerio*) line was used to assess cardiac function and blood flow. Zebrafish embryos were harvested from natural mating at the UCLA Zebrafish Core Facility, and they were maintained in the standard E3 medium and supplemented with phenylthiourea (PTU, 0.03%, Sigma Aldrich, MO) after 24 hours post fertilization (hpf) to inhibit melanogenesis. Embryos were anesthetized with tricaine (3-aminobenzoic acid ethyl ester, 0.2 mg/mL, Sigma Aldrich, MO), embedded in 1% low-melting-point agarose, and mounted in the Fluorinated Ethylene Propylene (FEP) under the microscope.

## Integration of light-sheet and light-field

We built an imaging setup with double-side illumination and dual-modality detection. Two dry objectives (Plan Fluor 4×/0.13, Nikon) were placed in the opposing direction. One generated a planar illumination with a cylindrical lens and the other one generated a rod-shaped volumetric illumination of controllable size with an aperture. They were perpendicularly positioned to the water dipping detection objective (Fluor 20×/0.5w, Nikon) and used for light-sheet and light-field imaging, respectively. For light-field detection, we used a macro lens (AF 60 mm 2.8D, Nikon) to relay the back focal plane of the microlens array (MLA, APO-Q-P150-F3.5 (633), OKO Optics) onto the camera sensor. The MLA was placed at the intermediate image plane. For both detections, sCMOS cameras (Flash 4.0 V2, Hamamatsu) were installed and synchronized by external trigger using LabVIEW. Zebrafish were held by a multi-dimension stage (Newport) to adjust their position and orientation and scanned by a stepper motorized actuator (ZST225B, Thorlabs) in the direction of the detection axis. Further information on the setup can be found in **S2 Fig**.

## Imaging pipeline and retrospective gating

Light-field and light-sheet imaging were sequentially applied with the retrospective gating method in the post-imaging processing for synchronization. During data capturing, both selective volume and selective plane illumination were employed. The light-field images of dsRed-labeled cells were recorded by light-field mode, while a 2-D cross-section of the GFP-labeled myocardium was concurrently captured by the light-sheet mode. The latter provided a time stamp to indicate the cardiac cycles of the acquired light fields. We further imaged the contracting myocardium in each 2-D slice throughout the entire 3-D heart in light-sheet mode. The scanning step size was 2 $\mu m$, and 50–80 image sequences were captured to cover the entire heart by the light-sheet imaging. Each image sequence, including light fields, time stamps, and light-sheet images, contained 450 frames to cover 4–5 cardiac cycles. Data were collected at the frame rate of 200 Hz (at a cropped frame size: 768 × 768 px) and exposure of 5 ms for all imaging tasks.

We adopted a deep-learning model to provide end-to-end conversion from the 2-D light field measurements to 3-D image stacks, providing a rapid 3-D reconstruction of the blood cells [31]. We trained the model with the static confocal images of blood cells in the hearts and vasculatures from 16 zebrafish embryos at 3–4 dpf. The trained model enabled the recovery of a 3-D volume (−76 $\mu m$ to 76 $\mu m$ in depth) from the light-field raw data. The final output was registered to the light sheet-acquired images of the contracting myocardium.

The synchronization that aligned all image sequences in identical cardiac phase was conceptually based on a previous retrospective gating method [7,20,21]. Assuming regular periodicity, we sought the proper relative temporal shifts between sequences based on the minimization of the difference function (for which we chose pixel-wise Euclidean distance).

The phase of light fields was indirectly synchronized via the time stamps. In the end, the temporally aligned data were merged for further quantification and visualization.

## Computation of vector mappings

We used an automatic tracking method to map the trajectories of blood cells in Imaris. The 3-D image sequence was resized to an isotropic spatial resolution of $2\,\mu m \times 2\,\mu m \times 2\,\mu m$, followed by a Gaussian blur to remove the image artifacts. Region of interest was restricted within the ventricle to decrease the computational cost. Point candidates were detected, and the trajectories were formed by an autoregressive motion algorithm with a maximum gap size of 1. We filtered the results using a threshold on track duration, and manual inspection was optional. Vector map indicating the transient motion of blood cells was computed from the trajectories in MATLAB.

The myocardial displacement was inferred by the intensity-based and non-rigid deformable image registration (DIR). We employed the demons method implemented in MATLAB's Image Processing Toolbox to estimate an optimal transformation, $T$: $(x, y, z) \rightarrow (x', y', z')$, mapping a voxel from the reference field to the moving field. We noted that image registration represents a geometric transformation of the image instead of an intensity transformation. In each voxel, a vector was predicted to represent the motion of the objects in the output transformation map, $T$. The 3-D images were resized to a spatial resolution of $2\,\mu m \times 2\,\mu m \times 2\,\mu m$. We removed the background by binary thresholding of the raw images. The ventricular region was defined in the surrounding area of the flow centerline. The registration was performed on 3 pyramid levels from coarse to fine resolution with 100, 50, and 25 iterations. To suppress the errors, we applied an average window of 7 frames (30 ms) to maintain smoothness between the successive maps.

## Quantification of the flow trajectories

To characterize the intracardiac flow pattern, we fitted a centerline to simplify the inward and outward geometry (S3 Fig). We collected the blood cells from the trajectories throughout multiple cardiac cycles. We performed a Principal Component Analysis (PCA) to find the principal plane of the points distribution and fitted a curve to these points in 3-D. We defined an angle $\theta$ for each point with respect to a pre-defined center point (red dots in S3B Fig) after projection onto the principal plane. This angle was used to define the orientation of the 3-D coordinates via the fourth polynomial functions by solving the least square problem. In parallel, a new coordinate system was defined at each point on the flow centerline where the tangent direction was used as the flow direction from AV canal to OFT (S3D Fig). We projected and averaged the vector maps onto the flow direction to obtain a collective flow velocity at each time point. The velocity distribution was visualized by collecting the maximum or average magnitudes at the specific positions (S4 Fig).

## Analysis of contractile function

The ventricle was divided into segments in reference to the blood flow centerline (S3D Fig). In this study, we analyzed 4 different segments of the myocardium in reference to the blood flow centerline (Figs 3 and 4), but the number of segments is freely defined by the user. In specific, the four segments were located on the left/right of the first half/last half of the flow centerline. The segmental velocity was represented by the mean velocity vector in the specific segment. The global velocity was computed as the average magnitude of the entire vector map.

## Supporting information

**S1 Fig. Pipeline for imaging and post-imaging processing.** The integration of light-field and light-sheet microscopy captures and synchronizes the blood flow and myocardial motion. Images of blood cells have been re-sampled to have 3-D isotropic spatial resolution, followed by Particle Tracking Velocimetry (PTV) to map the flow velocimetry. In parallel, deformable image registration (DIR) allows for displacement analysis of the myocardium. The flow center-line, inferred from blood cells distribution, also indicates the geometry of the ventricle; thereby, facilitating segmentation of the ventricular area for displacement analysis. The dual channel data are merged and visualized at the end.
(TIF)

**S2 Fig. A hybrid setup for light-field and light-sheet microscopy.** Schematic of the hybrid system. The laser with different wavelengths (473 nm and 532 nm) forms two pathways entering the opposing illumination objectives 1 & 2. Aperture, lens pairs and cylindrical lens are used to modulate the dimension and shape of the beam to form 1) the selective plane illumination for light-sheet microscopy and 2) the selective volume illumination for light-field microscopy. The fluorescent signal is collected by the detection lens orthogonal to the illumination. A dichromatic mirror (DM) partitions the detected fluorescent signal onto two detection modalities. For light-field detection, a microlens array (MLA) is placed on the intermediate image plane, and the sCMOS camera is conjugated to the back focal plane of MLA through a 1:1 relay lens.
(TIF)

**S3 Fig. Computation to generate the centerline for ventricular blood flow.** (A) Trajectories of the individual blood cells are positioned by automatic tracking throughout the cardiac cycle. (B) Principal Component Analysis (PCA) is used to define the main plane of the cell distribution (grey). The angle of each cell in reference to the center point (red) is defined as $\theta$. (C) For the PCA analysis, the covariance matrix is computed on the 3-D coordinates. By eigen value decomposition, we extract eigen vectors of the first two principal components to define the main plane. (D) A centerline is fitted through the blood cells. For each point along the center-line, we define a new coordinate system (see axis on the line). (E) Centerline fitting is derived as a least square problem. Each coordinate (x, y or z) of the point $X$ is fitted as the fourth order polynomial function with an angle $\theta$. Coefficients $P_1$, $P_2$, $P_3$ are computed by solving the minimization problem.
(TIF)

**S4 Fig. Velocity mapping for the individual blood cells.** (A) Mapping of the intracardiac blood cells with color-coded magnitude reveals the distribution of average velocity during the entire cardiac cycle. (B) Heatmaps are illustrated in the projected views. (C & D) The traveling cells are visualized in the transformed coordinate system with respect to the center flow line. The swapped coordinate system provides a standard perspective to analyze the intracardiac flow dynamics, bypassing the variations from the different imaging orientations. It also defines the positions of myocardial segments (**Figs 3 and 4**). (E) The elevated intracardiac flow velocity is highlighted in red, corresponding to the circled regions in (D).
(TIF)

**S5 Fig. Comparison between the deformation rates computed from ventricular kymograph and vector map by DIR.** (A) x-z cross-section of hearts at two time points. Dotted lines denote the sampling positions for kymographs. Scale bar, 30 $\mu m$. (B) Kymographs display the relative displacement of the ventricle throughout the cardiac cycles. (C) Distance between boundaries

are computed from kymographs. The derivative indicates the velocity of relative displacement, i.e. deformation rate. By sampling the heart in a direction (i) tilted from deformation direction (ii), one can enlarge the deformation rate drastically. Thus, kymograph based-analysis by 1D sampling is prone to angle-dependent variance. By sampling the vector map (D), a relative velocity can also be computed. In specific, the velocity is sampled at the intersection of the sampling line and the wall boundary. The relative velocity between two opposite intersection points represents the rate of relative displacement between two boundaries. It correlates with the result from properly sampled kymograph (ii) in term of magnitude and temporal pattern. (TIF)

**S6 Fig. Velocity gradient indicates the rate of local ventricular deformation.** (A) The velocity field illustrates ventricular contraction from the x-y and y-z views of the ventricle. (B) The velocity of ventricular contraction is depicted in the x, y and z axis direction. (C) The gradient of velocity in the x, y and z axis reveals the differential rates of regional/segmental wall deformation. (TIF)

**S7 Fig. Intracardiac flow domain from 2–3 dpf to 3–4 dpf.** Each figure was generated by the summation of different time points over the entire cardiac cycle. (A) Data were acquired during the early stage of cardiac looping (2–3 dpf) and during the (B) late stage (3–4 dpf). (TIF)

**S1 Movie. Dual-color 4-D beating zebrafish heart with flowing blood cells.** Two cardiac cycles have been shown. A, atrium. V, ventricle. Scale bar, 50 $\mu m$. (MOV)

**S2 Movie. Visualization of beating heart and blood flow in 3 example fish from 3 to 5 dpf.** Left to right, early to late stage. Myocardium is shown in green in x-y view. Flowing blood cells are shown in red in x-y and x-z views. Scale bar, 50 $\mu m$. (MOV)

**S3 Movie. Vector map of myocardial displacement and blood flow.** Blood cells have been tracked and each red vector demonstrates magnitude (vector length) and direction (vector direction) of the motion of each individual tracer. The black line is the flow centerline. Each green vector illustrates the magnitude (vector length) and direction (vector direction) of the displacement rate of the myocardium in corresponding voxel. Red vectors are mean vectors of the specific segment, after the heart is divided into 4 parts. Grid unit, $\mu m$. (MOV)

**S4 Movie. Visualization of beating heart and blood flow in 3 example fish at 3 dpf.** Myocardium is shown in green in x-y view. Flowing blood cells are shown in red in x-y and x-z views. Red arrow denotes the flow reversal. Scale bar, 50 $\mu m$. (MOV)

## Acknowledgments

We thank Yuan (Linda) Dong from UCLA zebrafish facility for her help in fish maintenance, Arthur Ko and Ryan O'Donnell for their generous help on manuscript proofreading.

## Author Contributions

**Conceptualization:** Zhaoqiang Wang, Tzung K. Hsiai.

**Data curation:** Zhaoqiang Wang.

**Formal analysis:** Zhaoqiang Wang.

**Funding acquisition:** Yichen Ding, Tzung K. Hsiai.

**Investigation:** Zhaoqiang Wang.

**Methodology:** Zhaoqiang Wang.

**Software:** Zhaoqiang Wang.

**Supervision:** Peng Fei, Tzung K. Hsiai.

**Validation:** Zhaoqiang Wang.

**Visualization:** Zhaoqiang Wang.

**Writing – original draft:** Zhaoqiang Wang.

**Writing – review & editing:** Zhaoqiang Wang, Yichen Ding, Sandro Satta, Mehrdad Roustaei, Peng Fei, Tzung K. Hsiai.

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
