## [Decision Letter · Decision Letter 0]

18 Nov 2020

Dear Dr. Hsiai,

Thank you very much for submitting your manuscript "A Hybrid of Light-Field and Light-Sheet Imaging to Decouple Myocardial Biomechanics from Intracardiac Flow Dynamics" for consideration at PLOS Computational Biology.

As with all papers reviewed by the journal, your manuscript was reviewed by members of the editorial board and by three independent reviewers. In light of the reviews (below this email), we would like to invite the resubmission of a significantly-revised version that takes into account the reviewers' comments.

In particular, the manuscript would benefit from a better summary of the prior art and its current major limitations. The analysis needs more validation and/or analysis of the variation between specimens. Please also discuss more clearly the major original contributions of the manuscript and their biological significance. Finally, one reviewer noted that the Github repository was not accessible.

We cannot make any decision about publication until we have seen the revised manuscript and your response to the reviewers' comments. Your revised manuscript is also likely to be sent to reviewers for further evaluation.

Sincerely,

Andrew D. McCulloch, Ph.D.

Associate Editor

PLOS Computational Biology

Daniel Beard

Deputy Editor

PLOS Computational Biology

Reviewer's Responses to Questions

**Comments to the Authors:**

Reviewer #1: Overall, this is a really interesting and important method to separate and measure the myocardial and hemodynamic kinematics of embryonic heart contractions, something that has typically been explored computationally or at a gross functional level. Understanding how contractility initiation and propagation corresponds with (and drives) local morphogenetic events is very important for understanding mechanisms of maturation and malformation. Although this current paper doesn't address either of those topics, the presented technique sets the stage for such questions to be answered in the zebrafish model to the degree that model system can recapitulate those events.

This being said, the current contributed data does not fully reflect the capability or potential of the measurement technique. The following concerns should be addressed to improve the validation and impact of the contribution:

1) The measurement approach appears to iterate through individual planar slabs of the myocardium. While this can create cubic voxels, it is unclear how accurate/precise the in-plane motions vs. out-of-plane deformations/velocities are within a specimen. Are these known? One simple way would be to re-analyze the same specimen by rotating either the scanning field or the specimen 90 degrees. Obtaining largely identical datasets would establish how locally variable that data is and whether it is biased to the plane of inquiry.

2) Related to #1, it is unclear how the local features measured in either tissue kinematics/velocities are shared in embryos of the same stage or across stages. This paper appears to measure the phenomena in an individual specimen, but does not compare across specimens for consistency across time/space. Shared features across embryos are more likely meaningful, as is an understanding of the variation present locally between embryos.

3) What is the developmental stage range that this technique can be used in the zebrafish? What are the failure modes that are encountered with respect to resolution, field of view, etc.? It doesn't appear that light penetration is the only limitation given the processing algorithms employed, but an understanding of these limitations would improve the proper use and interpretation of data from the approach.

Reviewer #2: This article describes a really nice technique to study the couple interaction of myocardial contraction and blood flow dynamics in the zebrafish heart (3 dpf). The technique combines light sheet microscopy and light field microscopy to simultaneously capture myocardial tissue contraction and red blood motion.

Comments and concerns (minor):

1. Deformable image registration, done to capture the myocardium dynamics, may not capture twisting dynamics of the heart tissue. Overall the image registration is best than nothing, but can be misleading too. Please acknowledge this in the discussion.

2. How was the 'retrospective' synchronization employed checked for accuracy?

3. After reading the description of the blood flow dynamics (results) I had the impression that there was no net flow, given that velocities were negative and negative displacement of red blood cells was about 50% (unless I misunderstood). It is mentioned that the dynamics is inefficient. However, when looking at the movies, the dynamics seem completely different. Blood flow advances and stops, but I cannot see significant backflow. What am I missing? Please describe the dynamics more accurately, or provide better movies.

4. Are there only very few red blood cells inside the ventricle? Is this accurate?

5. Comparing the tubular heart of the zebrafish heart analyzed to the mature heart of humans ('apex twisting') is not appropriate, I think. Heart structure is completely different, and while we can learn from the zebrafish, I feel this is not really a similarity to be drawn upon.

Reviewer #3: The review is uploaded as an attachment

**Have all data underlying the figures and results presented in the manuscript been provided?**

Reviewer #1: Yes

Reviewer #2: **No: **Other than code, the data is not available in a repository, it is states it is available upon request.

Reviewer #3: **No: **Github repository indicated by the authos (https://github.com/aaronzq/cardiac) was not accessible at the time of this review.

PLOS authors have the option to publish the peer review history of their article (what does this mean?). If published, this will include your full peer review and any attached files.

Reviewer #1: No

Reviewer #2: No

Reviewer #3: No
---

## [Decision Letter · Decision Letter 1]

8 Apr 2021

Dear Dr. Hsiai,

Thank you very much for submitting your manuscript "A Hybrid of Light-Field and Light-Sheet Imaging to Decouple Myocardial Biomechanics from Intracardiac Flow Dynamics" for consideration at PLOS Computational Biology. As with all papers reviewed by the journal, your manuscript was reviewed by members of the editorial board and by several independent reviewers. The reviewers appreciated the attention to an important topic. Based on the reviews, we are inviting you to modify the manuscript according to the review recommendations. To be acceptable for publication, the English language usage needs to be improved. I recommend using an experienced English language proof reader and editor. The reviewers also expressed concern that not all review comments were fully addressed and that the biomechanical significance of the new methods is difficult to assess.

Sincerely,

Andrew D. McCulloch, Ph.D.

Associate Editor

PLOS Computational Biology

Daniel Beard

Deputy Editor

PLOS Computational Biology

[LINK]

Reviewer's Responses to Questions

**Comments to the Authors:**

Reviewer #1: Overall the authors have been responsive to all of the reviewers comments, in particular with the generation of additional data with respect to multiple embryos to elaborate spatial/temporal variation in fluid and myocardial data. In their response to reviewers they acknowledge the challenge in scaling 3D/4D deformation (tissue strain) maps of the embryonic heart, but it is unclear whether those maps are indeed possible to be generated with the technology the authors have developed. It is unclear if their computational approach is capable of rendering fluid or solid biomechanical data as the title indicates. Fluid and tissue velocity profiles are interesting, but not the same as the functionally relevant tissue wall strain and shear stress. These parameters require reference frame decomposition and relative motion measurements between data points in the cloud, while single point velocity tracts (or averaging from a whole segment) does not. It is unclear to what degree of this biomechanically relevant data the authors claim their system is capable to do using the velocity data they currently have. If they indeed have point-to-point information of their solid boundaries and/or complete grids of fluid velocity data (as suggested in their vector maps), it should be straightforward to do. If not, how are their datasets digitized for analysis?

Reviewer #2: Concerns from reviewers 1 & 2 have mainly not been addressed. There is a response - but frequently not to the point and not directly addressing the reviewers concerns or aspects for clarification. As an example, the question of whether the retrospective synchronization was validated (reviewer 2, issue #2) is not addressed. I am guessing the answer is no. To validate you need to acquire images at different angles - including an angle that shows the wave propagation - the retrospective synchronization usually needs to be 'adjusted' with that information. A similar question, I believe, is asked by reviewer 1, (#1 and #2) and also not directly addressed.

Overall, however, answers to reviewer #3 seem to have been addressed. In particular the introduction and discussion, which have been completely re-written, seem to be much more focus on the actual contribution of the paper. These sections are now a really nice description of the methods employed and what is new in this paper - the combination of LSM and LSFM - to both capture myocardium motion and red blood cell velocities.

English needs significant revision - as several typos and grammatical issues can be found in the text. I would suggest that either the authors or the journal use an English Editing service before publication.

**Have all data underlying the figures and results presented in the manuscript been provided?**

Reviewer #1: Yes

PLOS authors have the option to publish the peer review history of their article (what does this mean?). If published, this will include your full peer review and any attached files.

Reviewer #1: No

Reviewer #2: No

**Have the authors made all data and (if applicable) computational code underlying the findings in their manuscript fully available?**

Reviewer #2: Yes

Figure Files:

Data Requirements:

Reproducibility:

References:

---

## [Decision Letter · Decision Letter 2]

11 Jun 2021

Dear Dr. Hsiai,

We are pleased to inform you that your manuscript 'A hybrid of light-field and light-sheet imaging to study myocardial function and intracardiac blood flow during zebrafish development' has been provisionally accepted for publication in PLOS Computational Biology.

Best regards,

Andrew D. McCulloch, Ph.D.

Associate Editor

PLOS Computational Biology

Daniel Beard

Deputy Editor

PLOS Computational Biology

Reviewer's Responses to Questions

**Comments to the Authors:**

Reviewer #1: The response to comments, textual changes, and additional data address my concerns.

Reviewer #2: Authors have addressed all remaining concerns.

**Have the authors made all data and (if applicable) computational code underlying the findings in their manuscript fully available?**

Reviewer #1: None

Reviewer #2: Yes

PLOS authors have the option to publish the peer review history of their article (what does this mean?). If published, this will include your full peer review and any attached files.

Reviewer #1: No

Reviewer #2: No

---

## [Editor Report · Acceptance letter]

30 Jun 2021

PCOMPBIOL-D-20-01650R2 

A hybrid of light-field and light-sheet imaging to study myocardial function and intracardiac blood flow during zebrafish development

Dear Dr Hsiai,

I am pleased to inform you that your manuscript has been formally accepted for publication in PLOS Computational Biology. Your manuscript is now with our production department and you will be notified of the publication date in due course.

With kind regards,

Olena Szabo
